# Linear convergence of the NQZ algorithm for finding the *H*-spectral radius of nonnegative tensors

Hongbin Lv[1]◉, Meixiang Chen ◉[2]¤◉*

1 Fujian Key Laboratory of Financial Information Processing, Putian University, Putian, Fujian, China,
2 Key Laboratory of Applied Mathematics of Fujian Province University, Putian University, Putian, Fujian, China

¤ Current address: School of Mathematics and Finance, Putian University, Putian, Fujian Province, China
◉ These authors contributed equally to this work.
* chenmx4406@ptu.edu.cn

## Abstract

The *R*-linear convergence of the NQZ algorithm for computing the *H*-spectral radius of a class of weakly irreducible nonnegative tensors is established by utilizing the directed graphs of tensors. Meanwhile, an upper bound for the root convergence factor *R* is derived and a general condition ensuring the linear convergence of the NQZ algorithm is provided.

## 1 Introduction

In recent years, tensor eigenvalue problems have gained increasing attention due to their broad applicability across various scientific and engineering domains. Notable examples include the best rank-one tensor approximation in data analytics [1,2], modeling of higher-order Markov chains [3], studies in solid mechanics and quantum entanglement [4,5], and structural analysis of multilayer networks [6,7]. The concept of tensor eigenvalues was independently introduced by Qi [8] and Lim [9] in 2005, marking a significant advance in tensor analysis. Building upon this foundation, Ng, Qi and Zhou [3] proposed the NQZ algorithm in 2009 to compute the *H*-spectral radius of irreducible nonnegative tensors. This algorithm serves as a fundamental tool in the fields of tensor spectral theory and numerical multilinear algebra, and is widely regarded as a natural extension of the classical power method for calculating the dominant eigenvalue of matrices. A substantial body of work has since been devoted to the analysis and development of the NQZ algorithm and its variants, leading to important theoretical and algorithmic advancements [3,10–17]. For example, Pearson [11] proved the convergence of the NQZ algorithm for essentially positive tensors. Furthermore, Liu, Zhou, and Ibrahim [13] examined a variant-referred to as the LZI algorithm and established its convergence for a class of primitive tensors.

Further progress has been made in understanding the convergence behavior of the NQZ algorithm under broader and more general conditions. In 2011, Chang,

**Data availability statement:** No datasets were generated during the current study.

**Funding:** This work was supported by the National Natural Science Foundation of China (Grant Nos. 61772292, 62372256; Principal Investigator: Meixiang Chen) and the Fujian Provincial Natural Science Foundation of China (Grant Nos. 2023J01997, 2024J01874; Principal Investigator: Hongbin Lv). The funders provided important support during the manuscript preparation and validation process.

**Competing interests:** The authors have declared that no competing interests exist.

Pearson, and Zhang [18] established the convergence of the NQZ algorithm for primitive tensors. Building on this work, Zhang and Qi [12] later proved the linear convergence of the NQZ algorithm for essentially positive tensors, and subsequently demonstrated the linear convergence of an improved variant known as the LZI algorithm for weakly positive tensors [14]. In particular, both analyses were performed under highly restrictive conditions, relying on the structural properties of the majorization matrix associated with a nonnegative tensor, which was required to be either a fully positive matrix or to have all positive off-diagonal entries. In 2014, Hu, Huang, and Qi [17] further advanced this line of research by proving the $R$-linear convergence of the NQZ algorithm for weakly primitive tensors. However, their analysis did not provide an explicit range or upper bound for the convergence factor $R$, limiting its practical applicability. Despite these constraints, these foundational works significantly enriched the theoretical development of the NQZ algorithm within the field of nonnegative tensor spectral analysis.

Subsequent studies have continued to broaden the scope of linear convergence analysis for algorithms computing the $H$-spectral radius of nonnegative tensors. For instance, in 2021, Zhang and Bu [16] introduced a diagonal similarity-based algorithm tailored for a newly defined class of weakly positive tensors and established its linear convergence. More recently, in 2024, Liu and Lv [10] extended the concepts of essentially positive, weakly positive, and generally weakly positive tensors by introducing the notion of weakly essentially irreducible nonnegative tensors. They proposed a corresponding algorithm for computing the $H$-spectral radius and established more general conditions ensuring its linear convergence. In parallel with these advances, alternative computational frameworks have also been developed. For example, Gautier, Tudisco, and Hein [19] introduced the concept of $(\sigma, p)$-eigenvalue for nonnegative tensors and designed a general algorithm applicable to weakly irreducible cases. In particular, when $\sigma = \{1, 2, \cdots, m\}, p = m$ the resulting $(\sigma, p)$-spectral radius coincides with the classical $H$-spectral radius, thus unifying and extending existing computational frameworks.

Despite these advances, the development of a unified and less restrictive convergence theory remains an open challenge. In this paper, we contribute to closing this gap by establishing the $R$-linear convergence of the NQZ algorithm from a novel perspective via the directed graph associated with a nonnegative tensor. Leveraging the structural characteristics of tensor-induced directed graphs, we not only derive an explicit upper bound for the convergence factor $R$, but also propose a more general sufficient condition for its linear convergence. This condition substantially relaxes the stringent assumptions imposed in previous works such as [12,14], thus broadening the theoretical foundation and enhancing the practical applicability of the NQZ algorithm.

This paper is organized as follows. Sect 2 provides a review of the relevant background and introduces the NQZ algorithm. In Sect 3, we establish the $R$-linear convergence of the NQZ algorithm for computing the $H$-spectral radius of a class of weakly irreducible nonnegative tensors by utilizing their associated directed graphs.

We also present more general conditions ensuring the linear convergence of the algorithm. Sect 4 summarizes the main findings of this work and discusses potential directions for future research.

## 2 Preliminaries

An $m$th-order tensor with dimension $n$ over the real numbers is a multi-way array consisting of $n^m$ real-valued entries, represented as

$$\mathcal{A} = (a_{i_1 i_2 \ldots i_m}), \ a_{i_1 i_2 \ldots i_m} \in \mathbb{R}, \text{for } i_j \in \langle n \rangle, \ j = 1, 2, \cdots, m,$$

where $\langle n \rangle = \{1, 2, \cdots, n\}$. In the special case when $m = 2$, the tensor reduces to an $n \times n$ matrix. A tensor $\mathcal{A}$ is termed nonnegative if every entry satisfies $a_{i_1 i_2 \ldots i_m} \geq 0$. The set of all real-valued tensors of order $m$ and dimension $n$ is denoted by $\mathbb{R}^{[m,n]}$, while $\mathbb{R}_+^{[m,n]}$ designates the subset comprising nonnegative tensors. Correspondingly, $\mathbb{R}^n$ denotes the space of all real $n$-dimensional vectors, with $\mathbb{R}_+^n$ and $\mathbb{R}_{++}^n$ representing the sets of nonnegative (including zero) and strictly positive vectors, respectively. Furthermore, $\mathbb{R}^{n \times n}$ denotes the set of all real $n \times n$ matrices.

In 2005, the concept of eigenvalues for tensors was independently introduced by Lim [8] and Qi [9].

**Definition 2.1.** *Consider a real tensor $\mathcal{A} = (a_{i_1 i_2 \ldots i_m})$ of order $m$ and dimension $n$, i.e., $\mathcal{A} \in \mathbb{R}^{[m,n]}$. A complex scalar $\lambda \in \mathbb{C}$ is called an* eigenvalue *of $\mathcal{A}$ if there exists a non-zero vector $\mathbf{x} = (x_1, x_2, \ldots, x_n)^\top \in \mathbb{C}^n$, such that*

$$\mathcal{A}\mathbf{x}^{m-1} = \lambda \mathbf{x}^{[m-1]},$$

*where the vector $\mathcal{A}\mathbf{x}^{m-1}$ is defined component-wise as*

$$(\mathcal{A}\mathbf{x}^{m-1})_i = \sum_{j_2, \ldots, j_m = 1}^{n} a_{ij_2 \ldots j_m} x_{j_2} \cdots x_{j_m}, \quad \text{for } i = 1, 2, \ldots, n,$$

*and the vector $\mathbf{x}^{[m-1]}$ is given by*

$$(\mathbf{x}^{[m-1]})_i = x_i^{m-1}, \quad \text{for } i = 1, 2, \ldots, n.$$

*In this setting, $\mathbf{x}$ is referred to as the* eigenvector *corresponding to the eigenvalue $\lambda$.*

*The* spectral radius *of the tensor $\mathcal{A}$, denoted by $\rho(\mathcal{A})$, is defined as the supremum of the absolute values of all its eigenvalues, i.e.,*

$$\rho(\mathcal{A}) = \sup\{|\lambda| : \lambda \in \text{spec}(\mathcal{A})\},$$

*where $\text{spec}(\mathcal{A})$ denotes the spectrum of $\mathcal{A}$.*

In particular, if both the eigenvalue $\lambda$ of $\mathcal{A}$ and its corresponding eigenvector $\mathbf{x}$ are real, i.e., $\lambda \in \mathbb{R}$ and $\mathbf{x} \in \mathbb{R}^n$, then the pair $(\lambda, \mathbf{x})$ is called an *H-eigenpair*, and $\lambda$ is referred to as an *H-eigenvalue* of $\mathcal{A}$.

In 2008, Chang et al. [20] extended the classical notion of irreducibility from matrices to tensors.

**Definition 2.2.** ([20]) *An $m$-th-order, $n$-dimensional tensor $\mathcal{A}$ is said to be* reducible *if there exists a nonempty proper subset $J \subset \langle n \rangle$ such that*

$$a_{i_1 i_2 \ldots i_m} = 0, \quad \text{for all } i_1 \in J, \text{ and } i_2, \ldots, i_m \notin J.$$

*If no such subset $J$ exists, then $\mathcal{A}$ is called* irreducible.

**Definition 2.3.** *Let $\mathcal{A} = (a_{i_1 i_2 \ldots i_m}) \in \mathbb{R}_+^{[m,n]}$. We recall several important notions related to $\mathcal{A}$:*

(1) ([11]) *The majorization matrix $M(\mathcal{A})$ associated with $\mathcal{A}$ is the nonnegative matrix whose (i,j)-th entry is given by*

$$(M(\mathcal{A}))_{ij} = a_{ij\ldots j}, \quad \text{for all } i,j \in \langle n \rangle.$$

(2) ([11,14]) *The tensor $\mathcal{A}$ is called essentially positive if $(M(\mathcal{A}))_{ij} > 0$ for every $i,j \in \langle n \rangle$. It is said to be weakly positive if $(M(\mathcal{A}))_{ij} > 0$ for all pairs $i \neq j$ with $i,j \in \langle n \rangle$.*

(3) ([16]) *The tensor $\mathcal{A}$ is termed generalized weakly positive if there exists an index $i_0 \in \langle n \rangle$ such that for all $j \in \langle n \rangle \setminus \{i_0\}$,*

$$a_{i_0 j \ldots j} > 0 \quad \text{and} \quad a_{j i_0 \ldots i_0} > 0.$$

The works presented in Refs [12,14,16] focus on the development of algorithms and the analysis of linear convergence for the *H*-spectral radius associated with essentially positive tensors, generalized weakly positive tensors, and weakly positive tensors, respectively. Moreover, in 2014, Hu et al. [17] introduced an equivalent characterization of weakly irreducible tensors, further enriching the theoretical framework in this area.

**Definition 2.4.** *Let $\mathcal{A} = (a_{i_1 i_2 \ldots i_m}) \in \mathbb{R}_+^{[m,n]}$ be a nonnegative tensor.*
(1) ([17]) *The representation matrix $G(\mathcal{A})$ associated with $\mathcal{A}$ is defined as the nonnegative matrix whose (i,j)-th entry is given by the sum of all entries $a_{i i_2 \ldots i_m}$ for which at least one of the indices $i_2, \ldots, i_m$ equals j.*
(2) ([17]) *The tensor $\mathcal{A}$ is said to be* weakly reducible *if the matrix $G(\mathcal{A})$ is reducible; otherwise, $\mathcal{A}$ is* weakly irreducible. *Furthermore, $\mathcal{A}$ is* weakly primitive *if $G(\mathcal{A})$ is a primitive matrix.*
(3) ([15]) *The tensor $\mathcal{A}$ is called* indirectly positive *if $G(\mathcal{A})$ is strictly positive, and* indirectly weakly positive *if $G(\mathcal{A}) + I$ is strictly positive, where I denotes the $n \times n$ identity matrix.*

Furthermore, Chang et al. [20] extended the classical Perron-Frobenius theorem from nonnegative matrices to nonnegative tensors.

**Theorem 2.1.** ([20]) *Let $\mathcal{A}$ be a nonnegative tensor of order m and dimension n. Then the following statements hold:*
(i) *There exist a scalar $\lambda_0 \geq 0$ and a nonnegative vector $\mathbf{x}_0 \in \mathbb{R}_+^n$ such that*

$$\mathcal{A}\mathbf{x}_0^{m-1} = \lambda_0 \mathbf{x}_0^{[m-1]}.$$

(ii) *If $\mathcal{A}$ is weakly irreducible, then $\lambda_0 > 0$ and the associated eigenvector $\mathbf{x}_0$ is strictly positive, that is, $\mathbf{x}_0 \in \mathbb{R}_{++}^n$. Moreover, $\lambda_0$ is the unique eigenvalue corresponding to a nonnegative eigenvector, and every eigenvalue $\lambda$ of $\mathcal{A}$ satisfies $|\lambda| \leq \lambda_0$.*

Based on statement (ii) of Theorem 2.1, it follows that the *H*-spectral radius of a nonnegative tensor is itself an eigenvalue and $\rho(\mathcal{A}) = \lambda_0$.

In 2010, Yang et al. [21] further generalized the classical bounds on the spectral radius from nonnegative matrices to nonnegative tensors.

**Theorem 2.2.** ([21]) *Let $\mathcal{A} = (a_{i_1 i_2 \ldots i_m}) \in \mathbb{R}_+^{[m,n]}$ be a nonnegative tensor and denote by $\rho(\mathcal{A})$ its H-spectral radius. Then the following inequalities hold:*

$$\min_{i \in \langle n \rangle} \sum_{i_2,\ldots,i_m=1}^{n} a_{i i_2 \ldots i_m} \leq \rho(\mathcal{A}) \leq \max_{i \in \langle n \rangle} \sum_{i_2,\ldots,i_m=1}^{n} a_{i i_2 \ldots i_m}.$$

Consider a nonnegative tensor $\mathcal{A} = (a_{i_1 \ldots i_m}) \in \mathbb{R}_+^{[m,n]}$. It can be represented by a directed graph $\mathbb{G}(\mathcal{A}) = (V, \mathbf{E}(\mathcal{A}))$, where the set of vertex is $V = \langle n \rangle$. A directed edge $(i,j)$ belongs to $\mathbf{E}(\mathcal{A})$ if there exist indices $\{i_2, \ldots, i_m\}$ such that $j \in \{i_2, \ldots, i_m\}$ and

the tensor entry $a_{ii_2\dots i_m}$ is non-zero. A walk from vertex $i$ to vertex $j$ in $\mathbb{G}(\mathcal{A})$ is a sequence of vertices $\gamma : i = \tilde{i}_0, \tilde{i}_1, \dots, \tilde{i}_r = j$ with each consecutive pair $(\tilde{i}_l, \tilde{i}_{l+1})$ being an edge in $\mathbf{E}(\mathcal{A})$ for $l = 0, 1, \dots, r-1$. If $i = j$, such a walk is called a non-simple path. When all vertices $\tilde{i}_0, \tilde{i}_1, \dots, \tilde{i}_r$ are distinct, $\gamma$ is referred to as a directed path connecting $i$ and $j$. The graph $\mathbb{G}(\mathcal{A})$ is said to be strongly connected if for every pair of distinct vertices $i$ and $j$, there exists a directed path from $i$ to $j$.

Based on this, Friedland et al. [22] introduced the concept of weak irreducibility for nonnegative tensors as follows:

**Definition 2.5.** ([22]) *An m-th order n-dimensional nonnegative tensor $\mathcal{A}$ is called weakly irreducible if its associated directed graph $\mathbb{G}(\mathcal{A})$ is strongly connected.*

In 2009, Ng et al. [3] proposed the NQZ method for the largest *H*-eigenvalue of a nonnegative irreducible tensor.

**Algorithm 1.** ([3]) **NQZ algorithm.**

**Step 0.** Choose $\mathbf{x}^{(0)} > 0, \mathbf{x}^{(0)} \in \mathbb{R}^n$. Let $\mathbf{y}^{(0)} = \mathcal{A}(\mathbf{x}^{(0)})^{m-1}$ and set $k := 0$.

**Step 1.** Compute

$$\mathbf{x}^{(k+1)} = \frac{(\mathbf{y}^{(k)})^{\left[\frac{1}{m-1}\right]}}{\|(\mathbf{y}^{(k)})^{\left[\frac{1}{m-1}\right]}\|}, \quad \mathbf{y}^{(k+1)} = \mathcal{A}(\mathbf{x}^{(k+1)})^{m-1},$$

$$\underline{\lambda}^{(k+1)} = \min_{x_i^{(k+1)} > 0} \frac{(\mathbf{y}^{(k+1)})_i}{(x_i^{(k+1)})^{m-1}}, \quad \bar{\lambda}^{(k+1)} = \max_{x_i^{(k+1)} > 0} \frac{(\mathbf{y}^{(k+1)})_i}{(x_i^{(k+1)})^{m-1}}.$$

**Step 2.** If $\bar{\lambda}^{(k+1)} = \underline{\lambda}^{(k+1)}$, stop. Otherwise, replace $k$ by $k+1$ and go to Step 1.

In this paper, for $\mathbf{x}, \mathbf{y} \in \mathbb{R}_{++}^n$, we define $\mathbf{x} \circ \mathbf{y} = (x_1 y_1, x_2 y_2, \cdots, x_n y_n)^{\mathsf{T}}$, $\frac{\mathbf{x}}{\mathbf{y}} = \{\frac{x_i}{y_i}\}$, $\mathbf{x}^{[\nu]} = (x_1^\nu, x_2^\nu, \cdots, x_n^\nu)^{\mathsf{T}}$, $\nu \in \mathbb{R}$, $\max\{\mathbf{x}\} = \max_{i \in \langle n \rangle} x_i$, $\min\{\mathbf{x}\} = \min_{i \in \langle n \rangle} x_i$, $\pi(i_2 \dots i_m)$ represents a permutation of the nodes $i_2, \dots, i_m$.

According to the NQZ algorithm, denoted as $\lambda_i^{(k)} = \frac{(\mathbf{y}^{(k)})_i}{((\mathbf{x}^{(k)})^{[m-1]})_i}$, $\lambda^{(k)} = (\lambda_1^{(k)}, \lambda_2^{(k)}, \dots, \lambda_n^{(k)})^{\mathsf{T}}$, we can obtain

$$\mathcal{A}(\mathbf{x}^{(k)})^{m-1} = \lambda^{(k)} \circ (\mathbf{x}^{(k)})^{[m-1]}.$$

## 3 Linear convergence of the NQZ algorithm

In this section, we establish the *R*-linear convergence of the NQZ algorithm by using the structural properties of directed graphs of tensors. Additionally, we derive an upper bound for the convergence factor $R$, which is associated with the directed paths of directed graphs of nonnegative tensors. Furthermore, we provide more generalized conditions for the linear convergence of the NQZ algorithm based on the parameters of the upper bound expression for $R$.

Let $\mathcal{A} = (a_{i_1 i_2 \dots i_m}) \in \mathbb{R}_+^{[m,n]}$. We define the set of all simple paths from node $i$ to node $j$ in the graph $\mathbb{G}(\mathcal{A})$ as

$$E_{i \to j}(\mathbb{G}(\mathcal{A})) = \{\gamma_{ij} | \gamma_{ij} : i = \tilde{i}_0 \to \tilde{i}_1 \to \tilde{i}_2 \to \cdots \to \tilde{i}_r = j, i \neq j\},$$

where $r_{ij} = |\gamma_{ij}| = r$ denotes the length of the path $\gamma_{ij} \in E_{i \to j}(\mathbb{G}(\mathcal{A}))$.

Additionally, define

$$r_{ji}^{\min}(\mathbb{G}(\mathcal{A})) = \min_{E_{j \to i}(\mathbb{G}(\mathcal{A}))} |\gamma_{ji}|,$$

and

$$r_i = \max_{j \in \langle n \rangle \setminus \{i\}} r_{ji}^{\min}(\mathbb{G}(\mathcal{A})).$$

Next, we demonstrate that during the computation of the *H*-spectral radius of a weakly irreducible nonnegative tensor using the NQZ algorithm, a consistent, nonzero, and nonnegative lower bound exists for all nonzero elements throughout the algorithm's iterations.

**Lemma 3.1.** ([3]) *Let* $\mathcal{A} = (a_{i_1 i_2 \cdots i_m}) \in \mathbb{R}_+^{[m,n]}$, $\rho(\mathcal{A})$ *be the H-spectral radius of* $\mathcal{A}$. *Then, we have* $\underline{\lambda}^{(k)}$ *monotonically increasing converges to* $\underline{\lambda} \geq 0$, $\overline{\lambda}^{(k)}$ *monotonically decreasing converges to* $\overline{\lambda}$, *and* $\underline{\lambda} \leq \rho(\mathcal{A}) \leq \overline{\lambda}$.

The proof process can be found in Theorem 2.4 of [3].

**Lemma 3.2.** *Let* $\mathcal{A} = (a_{i_1 i_2 \cdots i_m}) \in \mathbb{R}_+^{[m,n]}$ *be weakly irreducible. Then for the NQZ algorithm, we have*

$$\frac{\min\{\mathbf{x}^{(k)}\}}{\max\{\mathbf{x}^{(k)}\}} \geq \begin{cases} \left(\dfrac{\underline{a}}{\overline{\lambda}^{(0)}}\right)^{\frac{(m-1)^{n-1}-1}{m-2}} > 0, & m \geq 3, \\[2ex] \left(\dfrac{\underline{a}}{\overline{\lambda}^{(0)}}\right)^{n-1} > 0, & m = 2, \end{cases} \tag{1}$$

*where* $\underline{a} = \min\{a_{i_1 i_2 \cdots i_m} > 0 : i_1, i_2, \cdots, i_m \in \langle n \rangle\}$, $\overline{\lambda}^{(0)} = \max_{i \in \langle n \rangle} \sum_{i_2, \cdots, i_m = 1}^{n} a_{i i_2 \cdots i_m}$.

Proof.    By displacement algorithm, we know $\mathbf{x}^{(k)} = (x_1^{(k)}, x_2^{(k)}, \cdots, x_n^{(k)})^{\mathsf{T}} \in \mathbb{R}_{++}^n$. Without loss of generality, assume

$$x_{t_n}^{(k)} \geq x_{t_{n-1}}^{(k)} \geq \cdots \geq x_{t_2}^{(k)} \geq x_{t_1}^{(k)} > 0. \tag{2}$$

Since the tensor $\mathcal{A}$ is weakly irreducible, according to Definition 2.2, there exists a directed path $\gamma$ from vertex $t_1$ to vertex $t_n$ in the associated digraph $\mathbb{G}(\mathcal{A})$, where $\gamma : t_1 = \tilde{i}_0 \to \tilde{i}_1 \to \tilde{i}_2 \to \cdots \to \tilde{i}_{r-1} \to \tilde{i}_r = t_n, \tilde{i}_0, \tilde{i}_1, \tilde{i}_2, \cdots, \tilde{i}_{r-1}, \tilde{i}_r \in \langle n \rangle, 1 \leq r \leq n - 1$, such that $a_{t_1 \pi(i_2 \cdots \tilde{i}_1 \cdots i_m)} > 0$, $a_{\tilde{i}_1 \pi(i_2 \cdots \tilde{i}_2 \cdots i_m)} > 0$, $\cdots$, $a_{\tilde{i}_{r-1} \pi(i_2 \cdots \tilde{i}_r \cdots i_m)} > 0$. Therefore, combining Lemma 3.1, (2) and $k \in \mathbb{Z}^+ \cup \{0\}$, we have

$$\overline{\lambda}^{(0)} \left(x_{t_1}^{(k)}\right)^{m-1} \geq \overline{\lambda}^{(k)} \left(x_{t_1}^{(k)}\right)^{m-1} \geq \lambda_{t_1}^{(k)} \left(x_{t_1}^{(k)}\right)^{m-1}$$

$$= \sum_{i_2, \cdots, i_m = 1}^{n} a_{t_1 i_2 \cdots i_m} x_{i_2}^{(k)} \cdots x_{i_m}^{(k)}$$

$$\geq a_{t_1 \pi(i_2 \cdots \tilde{i}_1 \cdots i_m)} x_{i_2}^{(k)} \cdots x_{\tilde{i}_1}^{(k)} \cdots x_{i_m}^{(k)}$$

$$\geq \underline{a} \left(x_{t_1}^{(k)}\right)^{m-2} x_{\tilde{i}_1}^{(k)},$$

i.e.,

$$\overline{\lambda}^{(0)} \cdot x_{t_1}^{(k)} \geq \underline{a} \cdot x_{\tilde{i}_1}^{(k)}. \tag{3}$$

Similarly, we get

$$\overline{\lambda}^{(0)}\left(x_{\tilde{i}_1}^{(k)}\right)^{m-1} \geq \underline{a}\left(x_{t_1}^{(k)}\right)^{m-2} x_{\tilde{i}_2}^{(k)}, \tag{4}$$

$$\overline{\lambda}^{(0)}\left(x_{\tilde{i}_2}^{(k)}\right)^{m-1} \geq \underline{a}\left(x_{t_1}^{(k)}\right)^{m-2} x_{\tilde{i}_3}^{(k)}, \tag{5}$$

$$\cdots,$$

$$\overline{\lambda}^{(0)}\left(x_{\tilde{i}_{r-1}}^{(k)}\right)^{m-1} \geq \underline{a}\left(x_{t_1}^{(k)}\right)^{m-2} x_{t_n}^{(k)}. \tag{6}$$

So by (3) and (4), we obtain

$$\left(\overline{\lambda}^{(0)} \cdot x_{t_1}^{(k)}\right)^{m-1} \cdot \overline{\lambda}^{(0)}\left(x_{\tilde{i}_1}^{(k)}\right)^{m-1} \geq \left(\underline{a} \cdot x_{\tilde{i}_1}^{(k)}\right)^{m-1} \cdot \underline{a}\left(x_{t_1}^{(k)}\right)^{m-2} x_{\tilde{i}_2}^{(k)},$$

that is,

$$(\overline{\lambda}^{(0)})^{(m-1)+1} \cdot x_{t_1}^{(k)} \geq \underline{a}^{(m-1)+1} \cdot x_{\tilde{i}_2}^{(k)}. \tag{7}$$

By (7) and (5), we can get

$$\left((\overline{\lambda}^{(0)})^{(m-1)+1} \cdot x_{t_1}^{(k)}\right)^{m-1} \cdot \overline{\lambda}^{(0)}\left(x_{\tilde{i}_2}^{(k)}\right)^{m-1} \geq \left(\underline{a}^{(m-1)+1} \cdot x_{\tilde{i}_2}^{(k)}\right)^{m-1} \cdot \underline{a}\left(x_{t_1}^{(k)}\right)^{m-2} x_{\tilde{i}_3}^{(k)},$$

that is,

$$(\overline{\lambda}^{(0)})^{(m-1)^2+(m-1)+1} \cdot x_{t_1}^{(k)} \geq \underline{a}^{(m-1)^2+(m-1)+1} \cdot x_{\tilde{i}_3}^{(k)}.$$

Following this sequence of steps, we obtain

$$(\overline{\lambda}^{(0)})^{(m-1)^{r-1}+(m-1)^{r-2}+\cdots+(m-1)+1} \cdot x_{t_1}^{(k)} \geq \underline{a}^{(m-1)^{r-1}+(m-1)^{r-2}+\cdots+(m-1)+1} \cdot x_{t_n}^{(k)}. \tag{8}$$

When $m \geq 3$, it can be obtained from equation (8) that

$$(\overline{\lambda}^{(0)})^{\frac{(m-1)^r-1}{m-2}} \cdot x_{t_1}^{(k)} \geq \underline{a}^{\frac{(m-1)^r-1}{m-2}} \cdot x_{t_n}^{(k)},$$

that is,

$$\frac{\min\{\mathbf{x}^{(k)}\}}{\max\{\mathbf{x}^{(k)}\}} = \frac{x_{t_1}^{(k)}}{x_{t_n}^{(k)}} \geq \left(\frac{\underline{a}}{\bar{\lambda}^{(0)}}\right)^{\frac{(m-1)^r-1}{m-2}} \geq \left(\frac{\underline{a}}{\bar{\lambda}^{(0)}}\right)^{\frac{(m-1)^{n-1}-1}{m-2}} > 0.$$

When $m = 2$, it can be obtained from equation (8) that

$$\frac{\min\{\mathbf{x}^{(k)}\}}{\max\{\mathbf{x}^{(k)}\}} = \frac{x_{t_1}^{(k)}}{x_{t_n}^{(k)}} \geq \left(\frac{\underline{a}}{\bar{\lambda}^{(0)}}\right)^r \geq \left(\frac{\underline{a}}{\bar{\lambda}^{(0)}}\right)^{n-1} > 0.$$

This completes the proof.

In 2014, Hu et al. [17] proved the *R*-linear convergence of *H*-spectral radius NQZ algorithms for weakly primitive tensors. In contrast to their work, we establish the *R*-linear convergence of the NQZ algorithm for a class of weakly irreducible nonnegative tensors from a different perspective, employing directed graphs of tensors. Additionally, we derive an upper bound on the root convergence factor *R* and provide a general condition for the linear convergence of the NQZ algorithm.

**Theorem 3.1.** *Let* $\mathcal{A} = (a_{i_1 i_2 \ldots i_m}) \in \mathbb{R}_+^{[m,n]}$ *be weakly irreducible. Define*

$$r_0 = \min_{\substack{i \in \langle n \rangle \\ e_{ii} \in \mathbb{G}(\mathcal{A})}} \max_{j \in \langle n \rangle \setminus \{i\}} r_{ji}^{\min}(\mathbb{G}(\mathcal{A})),$$

*if there exists* $i_0 \in \langle n \rangle$ *such that* $e_{i_0 i_0} \in \mathbb{G}(\mathcal{A})$ *then for the NQZ algorithm, when* $k \geq 1$, *the following inequality holds:*

$$\bar{\lambda}^{((k+1)r_0)} - \underline{\lambda}^{((k+1)r_0)} \leq \alpha(\bar{\lambda}^{(kr_0)} - \underline{\lambda}^{(kr_0)}) \leq \alpha^{k+1}(\bar{\lambda}^{(0)} - \underline{\lambda}^{(0)}),$$

*Furthermore, for* $l \geq r_0 + 1$,

$$\bar{\lambda}^{(l)} - \underline{\lambda}^{(l)} \leq \alpha^{l-r_0}\left(\bar{\lambda}^{(0)} - \underline{\lambda}^{(0)}\right),$$

*it follows that the NQZ algorithm exhibits R-linear convergence, where the convergence factor* $\alpha$ *satisfies* $0 < \alpha < 1, 0 < \gamma' < 1$, *and*

$$\alpha = \begin{cases} 1 - \gamma'\left(\left(\frac{\underline{a}}{\bar{\lambda}^{(0)}}\right)^{\frac{(m-1)^n-(m-1)}{m-2}+1} \cdot \frac{1}{m-1}\right)^{r_0}, & m \geq 3, \\ 1 - \gamma'\left(\left(\frac{\underline{a}}{\bar{\lambda}^{(0)}}\right)^n\right)^{r_0}, & m = 2. \end{cases} \quad (9)$$

*The value of* $\gamma'$ *is shown in the proof.*

Proof.    Only the proof for $m \geq 3$ is given, and the proof for $m = 2$ is similar.

Assuming $\underline{\lambda}^{(0)} < \bar{\lambda}^{(0)}$, otherwise if $\underline{\lambda}^{(0)} = \bar{\lambda}^{(0)}$, we can obtain $\underline{\lambda}^{(0)} \leq \rho(\mathcal{A}) \leq \bar{\lambda}^{(0)}$ from Theorem 2.2, and then $\underline{\lambda}^{(0)} = \rho(\mathcal{A}) = \bar{\lambda}^{(0)}$.

From the NQZ algorithm, it follows that

$$\lambda^{(k+1)} = \frac{\mathcal{A}(\mathbf{x}^{(k+1)})^{m-1}}{(\mathbf{x}^{(k+1)})^{[m-1]}}$$

$$= \frac{\mathcal{A}\left(\left(\left(\mathcal{A}(\mathbf{x}^{(k)})^{m-1}\right)^{\left[\frac{1}{m-1}\right]}\right)\right)^{m-1}}{\left(\left(\mathcal{A}(\mathbf{x}^{(k)})^{m-1}\right)^{\left[\frac{1}{m-1}\right]}\right)^{[m-1]}}$$

$$= \frac{\mathcal{A}\left(\left(\left(\lambda^{(k)} \circ \left(\mathbf{x}^{(k)}\right)^{[m-1]}\right)^{\left[\frac{1}{m-1}\right]}\right)\right)^{m-1}}{\left(\left(\lambda^{(k)} \circ \left(\mathbf{x}^{(k)}\right)^{[m-1]}\right)^{\left[\frac{1}{m-1}\right]}\right)^{[m-1]}}$$

$$= \frac{\mathcal{A}\left(\left(\lambda^{(k)}\right)^{\left[\frac{1}{m-1}\right]} \circ \mathbf{x}^{(k)}\right)^{m-1}}{\lambda^{(k)} \circ \left(\mathbf{x}^{(k)}\right)^{[m-1]}}. \tag{10}$$

(I) Take $i_0 \in E_0 = \{i \in \langle n \rangle : \min\limits_{\substack{i \in \langle n \rangle \\ e_{ii} \in \mathbb{G}(\mathcal{A})}} \max\limits_{j \in \langle n \rangle \setminus \{i\}} r_{ji}^{\min}(\mathbb{G}(\mathcal{A}))\}$, and assume that $\underline{\lambda}^{(0)} < \lambda_{i_0}^{(0)} < \bar{\lambda}^{(0)}$.

(i) Since $e_{i_0 i_0} \in \mathbb{G}(\mathcal{A})$, there exists a permutation $\pi(i_2, \cdots, i_0, \cdots, i_m)$ such that $a_{i_0 \pi(i_2 \cdots i_0 \cdots i_m)} > 0$. For $i_0$, by Equation (10), we obtain

$$\lambda_{i_0}^{(1)} = \bar{\lambda}^{(0)} - \frac{\sum\limits_{i_2, \cdots, i_m = 1}^{n} a_{i_0 i_2 \cdots i_m} \left[\bar{\lambda}^{(0)} - \left(\lambda_{i_2}^{(0)} \cdots \lambda_{i_m}^{(0)}\right)^{\frac{1}{m-1}}\right] x_{i_2}^{(0)} \cdots x_{i_m}^{(0)}}{\lambda_{i_0}^{(0)} \cdot \left(x_{i_0}^{(0)}\right)^{m-1}}$$

$$\leq \bar{\lambda}^{(0)} - a_{i_0 \pi(i_2 \cdots i_0 \cdots i_m)} \frac{\bar{\lambda}^{(0)} - \left(\bar{\lambda}^{(0)}\right)^{\frac{m-2}{m-1}} \left(\lambda_{i_0}^{(0)}\right)^{\frac{1}{m-1}}}{\lambda_{i_0}^{(0)}} \cdot \left(\frac{a}{\bar{\lambda}^{(0)}}\right)^{\frac{(m-1)^n - (m-1)}{m-2}}$$

$$\leq \bar{\lambda}^{(0)} - \left(\frac{a}{\bar{\lambda}^{(0)}}\right)^{\frac{(m-1)^n - (m-1)}{m-2} + 1} \cdot \frac{1}{m-1} \cdot \left(\bar{\lambda}^{(0)} - \lambda_{i_0}^{(0)}\right)$$

$$\leq \bar{\lambda}^{(0)} - \gamma \cdot \left(\frac{a}{\bar{\lambda}^{(0)}}\right)^{\frac{(m-1)^n - (m-1)}{m-2} + 1} \cdot \frac{1}{m-1} \cdot \left(\bar{\lambda}^{(0)} - \underline{\lambda}^{(0)}\right)$$

$$= \bar{\lambda}^{(0)} - \gamma \delta \left(\bar{\lambda}^{(0)} - \underline{\lambda}^{(0)}\right),$$

where $0 < \gamma < 1$, satisfying $\bar{\lambda}^{(0)} - \lambda_{i_0}^{(0)} > \gamma(\bar{\lambda}^{(0)} - \underline{\lambda}^{(0)})$, $0 < \delta = \left(\frac{a}{\bar{\lambda}^{(0)}}\right)^{\frac{(m-1)^n - (m-1)}{m-2} + 1} \cdot \frac{1}{m-1} < 1$.

Applying the above equation, a similar result can be obtained

$$\lambda_{i_0}^{(2)} \leq \overline{\lambda}^{(0)} - \gamma\delta^2(\overline{\lambda}^{(0)} - \underline{\lambda}^{(0)}).$$

And then, similarly to the discussion above, we have

$$\lambda_{i_0}^{(t)} \leq \overline{\lambda}^{(0)} - \gamma\delta^t(\overline{\lambda}^{(0)} - \underline{\lambda}^{(0)}) , t = 1, 2, \cdots, r_0.$$

Similar, when $\overline{\lambda}^{(r_0)} - \lambda_{i_0}^{(r_0)} \geq \gamma(\overline{\lambda}^{(r_0)} - \underline{\lambda}^{(r_0)})$, then we have

$$\lambda_{i_0}^{(r_0+t)} \leq \overline{\lambda}^{(r_0)} - \gamma\delta^t(\overline{\lambda}^{(r_0)} - \underline{\lambda}^{(r_0)}) , t = 1, \cdots, r_0.$$

When $\overline{\lambda}^{(r_0)} - \lambda_{i_0}^{(r_0)} < \gamma(\overline{\lambda}^{(r_0)} - \underline{\lambda}^{(r_0)})$, then we obtain

$$\lambda_{i_0}^{(r_0)} - \underline{\lambda}^{(r_0)} > (1 - \gamma)(\overline{\lambda}^{(r_0)} - \underline{\lambda}^{(r_0)}),$$

a similar discussion leads to

$$\lambda_{i_0}^{(r_0+t)} \geq \underline{\lambda}^{(r_0)} + (1 - \gamma)\delta^t \left( \overline{\lambda}^{(r_0)} - \underline{\lambda}^{(r_0)} \right), \ t = 1, 2, \cdots, r_0.$$

(ii) For any $i \in \langle n \rangle, i \neq i_0$, since $\mathcal{A}$ is weakly irreducible, we know that $\mathbb{G}(\mathcal{A})$ is strongly connected, and so there exist $a_{\tilde{i}_1\pi(i_2\cdots i_0\cdots i_m)} > 0, a_{\tilde{i}_2\pi(i_2\cdots\tilde{i}_1\cdots i_m)} > 0, \cdots, a_{\tilde{i}_{r-1}\pi(i_2\cdots\tilde{i}_{r-2}\cdots i_m)} > 0, a_{i\pi(i_2\cdots\tilde{i}_{r-1}\cdots i_m)} > 0$, where $i_0, \tilde{i}_1, \cdots, \tilde{i}_{r-1}, i$ are not the same as each other, and $r \leq r_0$. There are two cases:

(1) When $\overline{\lambda}^{(kr_0)} - \lambda_{i_0}^{(kr_0)} \geq \gamma(\overline{\lambda}^{(kr_0)} - \underline{\lambda}^{(kr_0)})$, we have

$$\begin{aligned}
\lambda_{\tilde{i}_1}^{(kr_0+1)} &\leq \overline{\lambda}^{(kr_0)} - a_{\tilde{i}_1\pi(i_2\cdots i_0\cdots i_m)} \\
&\cdot \frac{\left( \overline{\lambda}^{(kr_0)} - \prod\limits_{l\in\{i_2\cdots i_0\cdots i_m\}} \left( \lambda_l^{(kr_0)} \right)^{\frac{1}{m-1}} \right) \prod\limits_{l\in\{i_2\cdots i_0\cdots i_m\}} \left( x_l^{(kr_0)} \right)}{\lambda_{\tilde{i}_1}^{(kr_0)} \left( x_{\tilde{i}_1}^{(kr_0)} \right)^{m-1}} \\
&\leq \overline{\lambda}^{(kr_0)} - \left( \frac{a}{\overline{\lambda}^{(0)}} \right)^{\frac{(m-1)^n - (m-1)}{m-2} + 1} \cdot \frac{1}{m-1} \cdot \left( \overline{\lambda}^{(kr_0)} - \lambda_{i_0}^{(kr_0)} \right) \\
&< \overline{\lambda}^{(kr_0)} - \gamma\delta \left( \overline{\lambda}^{(kr_0)} - \underline{\lambda}^{(kr_0)} \right).
\end{aligned}$$

By applying the previous equation in sequence, we ultimately obtain

$$\lambda_i^{((k+1)r_0)} < \overline{\lambda}^{(kr_0)} - \gamma\delta^{r_0} \left( \overline{\lambda}^{(kr_0)} - \underline{\lambda}^{(kr_0)} \right).$$

(2) When $\overline{\lambda}^{(kr_0)} - \lambda_{i_0}^{(kr_0)} < \gamma(\overline{\lambda}^{(kr_0)} - \underline{\lambda}^{(kr_0)})$, a discussion similar to (1) leads to

$$\lambda_i^{((k+1)r_0)} \geq \underline{\lambda}^{(kr_0)} + (1-\gamma)\delta^{r_0}\left(\overline{\lambda}^{(kr_0)} - \underline{\lambda}^{(kr_0)}\right).$$

Combining (i) and (ii) by Lemma 3.1, for any $k \in \mathbb{Z}^+ \cup \{0\}$, there are always

$$\overline{\lambda}^{((k+1)r_0)} - \underline{\lambda}^{((k+1)r_0)} \leq \overline{\lambda}^{((k+1)r_0)} - \underline{\lambda}^{(kr_0)} \leq (1-\gamma\delta^{r_0})\left(\overline{\lambda}^{(kr_0)} - \underline{\lambda}^{(kr_0)}\right),$$

or

$$\overline{\lambda}^{((k+1)r_0)} - \underline{\lambda}^{((k+1)r_0)} \leq \overline{\lambda}^{(kr_0)} - \underline{\lambda}^{((k+1)r_0)} \leq (1-(1-\gamma)\delta^{r_0})\left(\overline{\lambda}^{(kr_0)} - \underline{\lambda}^{(kr_0)}\right).$$

Take $\gamma' = \min\{\gamma, 1-\gamma\}$, then there are

$$\overline{\lambda}^{((k+1)r_0)} - \underline{\lambda}^{((k+1)r_0)} \leq (1-\gamma'\delta^{r_0})\left(\overline{\lambda}^{(kr_0)} - \underline{\lambda}^{(kr_0)}\right)$$

$$\leq \cdots \leq (1-\gamma'\delta^{r_0})^{k+1}\left(\overline{\lambda}^{(0)} - \underline{\lambda}^{(0)}\right).$$

Denote $\alpha = 1 - \gamma'\delta^{r_0}$, then $0 < \alpha < 1$, there are

$$\overline{\lambda}^{(kr_0)} - \underline{\lambda}^{(kr_0)} \leq \alpha^k\left(\overline{\lambda}^{(0)} - \underline{\lambda}^{(0)}\right). \tag{11}$$

For any $l \geq r_0 + 1$, there exists a positive integer $k \in \mathbb{Z}^+$ such that $(k-1)r_0 \leq l < kr_0$. This implies that $k > \frac{l}{r_0}$, by (7), we have

$$\overline{\lambda}^{(l)} - \underline{\lambda}^{(l)} \leq \alpha^{(k-1)r_0}\left(\overline{\lambda}^{(0)} - \underline{\lambda}^{(0)}\right) \leq \alpha^{l-r_0}\left(\overline{\lambda}^{(0)} - \underline{\lambda}^{(0)}\right).$$

By

$$R = \limsup_{l\to\infty}(\overline{\lambda}^{(l)} - \underline{\lambda}^{(l)})^{\frac{1}{l}} \leq \limsup_{l\to\infty}\left(\alpha^{l-r_0}\left(\overline{\lambda}^{(0)} - \underline{\lambda}^{(0)}\right)\right)^{\frac{1}{l}} = \alpha$$

we know that $0 < R < 1$. Therefore, the NQZ algorithm is $R$-linearly convergent, and the root convergence factor $R \leq \alpha$.

Applying Lemma 3.1 there are

$$\overline{\lambda} - \underline{\lambda} = \lim_{k\to\infty}\left(\overline{\lambda}^{(kr_0)} - \underline{\lambda}^{(kr_0)}\right) \leq \lim_{k\to\infty}\alpha^k\left(\overline{\lambda}^{(0)} - \underline{\lambda}^{(0)}\right) = 0,$$

applying Theorem 2.1 again we have $\overline{\lambda} = \underline{\lambda} = \rho(\mathcal{A})$.

(II) When $\lambda_{i_0}^{(0)} = \overline{\lambda}^{(0)}$, or $\lambda_{i_0}^{(0)} = \underline{\lambda}^{(0)}$, assume that $\lambda_{i_0}^{(0)} = \overline{\lambda}^{(0)}$, then we have

$$\lambda_{i_0}^{(1)} \le \overline{\lambda}^{(0)} - a_{i_0\pi(i_2\cdots i_0\cdots i_m)} \frac{\overline{\lambda}^{(0)} - \left(\overline{\lambda}^{(0)}\right)^{\frac{m-2}{m-1}} \left(\lambda_{i_0}^{(0)}\right)^{\frac{1}{m-1}}}{\lambda_{i_0}^{(0)}} \cdot \left(\frac{\underline{a}}{\overline{\lambda}^{(0)}}\right)^{\frac{(m-1)^n-(m-1)}{m-2}}$$

$$\le \overline{\lambda}^{(0)} - \left(\frac{\underline{a}}{\overline{\lambda}^{(0)}}\right)^{\frac{(m-1)^n-(m-1)}{m-2}+1} \cdot \frac{1}{m-1} \cdot \left(\overline{\lambda}^{(0)} - \lambda_{i_0}^{(0)}\right)$$

$$= \overline{\lambda}^{(0)} - \left(\frac{\underline{a}}{\overline{\lambda}^{(0)}}\right)^{\frac{(m-1)^n-(m-1)}{m-2}+1} \cdot \frac{1}{m-1} \cdot \left(\overline{\lambda}^{(0)} - \underline{\lambda}^{(0)}\right)$$

$$< \overline{\lambda}^{(0)} - \gamma\delta\left(\overline{\lambda}^{(0)} - \underline{\lambda}^{(0)}\right),$$

where $0 < \gamma < 1$.

Thus, a discussion similar to that of (I) leads to the conclusion.

**Remark 3.1.** *Using Theorem 3.1, we can prove that the NQZ algorithm is R-linearly convergent and provide an upper bound on its root convergence factor, which. is $0 < R \le \alpha$, where the value of $\alpha$ is shown in equation (9).*

**Remark 3.2.** *For a general weakly irreducible nonnegative tensor $\mathcal{A}$, it suffices to define $\mathcal{B} = \mathcal{A} + \mu\mathcal{E}$, where $\mu > 0$. This ensures that $\mathcal{B}$ satisfies the conditions of Theorem 3.1.*

Applying Theorem 3.1, we obtain

**Theorem 3.2.** *Let $\mathcal{A} = (a_{i_1i_2\cdots i_m}) \in \mathbb{R}_+^{[m,n]}$ be weakly irreducible. If for any $i \in \langle n \rangle$, it holds that $e_{ii} \in \mathbb{G}(\mathcal{A})$, then for a given $0 < \varepsilon < \overline{\lambda}^{(0)} - \underline{\lambda}^{(0)}$, when $k > \frac{\ln\varepsilon - \ln(\overline{\lambda}^{(0)} - \underline{\lambda}^{(0)})}{\ln\alpha} \cdot r_0$, applying the NQZ algorithm, there must be $\overline{\lambda}^{(k)} - \underline{\lambda}^{(k)} < \varepsilon$.*

More general conditions for the linear convergence of the NQZ algorithm can be easily derived from Theorem 3.1.

**Theorem 3.3.** *Let $\mathcal{A} = (a_{i_1i_2\cdots i_m}) \in \mathbb{R}_+^{[m,n]}$ be weakly irreducible. If there exists an index $i_0 \in \langle n \rangle$, such that $e_{i_0i_0} \in \mathbb{G}(\mathcal{A})$, and $e_{ii_0} \in \mathbb{G}(\mathcal{A})$ for any $i \in \langle n \rangle \backslash \{i_0\}$, then by applying the NQZ algorithm,*

$$\overline{\lambda}^{(k+1)} - \underline{\lambda}^{(k+1)} \le \alpha\left(\overline{\lambda}^{(k)} - \underline{\lambda}^{(k)}\right), \ k \in \mathbb{Z}^+ \cup \{0\},$$

*where the value of $0 < \alpha < 1$ is given in Theorem 3.1.*

Proof.    Since $e_{ii_0} \in \mathbb{G}(\mathcal{A})$ for any $i \in \langle n \rangle \backslash \{i_0\}$, we have $\max\limits_{j\in\langle n\rangle\backslash\{i_0\}} r_{ji_0}^{\min}(\mathbb{G}(\mathcal{A})) = 1$, therefore,

$$r_0 = \min\limits_{\substack{i\in\langle n\rangle \\ e_{ii}\in\mathbb{G}(\mathcal{A})}} \max\limits_{j\in\langle n\rangle\backslash\{i\}} r_{ji}^{\min}(\mathbb{G}(\mathcal{A})) = 1,$$

Thus, the conclusion follows from Theorem 3.1 and its proof.

**Remark 3.3.** *Theorem 3.2 establishes more general conditions for the linear convergence of the NQZ algorithm in computing the H-spectral radius of a nonnegative tensor, based on the directed graph associated with the tensor.*

Literature in Refs [12,14] demonstrates that the NQZ algorithm achieves linear convergence when computing the $H$-spectral radius of essentially positive tensors or weakly positive tensors. The following example demonstrates that the condition proposed in this article for the linear convergence of the NQZ algorithm, as stated in Theorem 3.2, provides a more comprehensive and generalized framework compared to the results presented in [12,14].

**Example 3.1.** *Let $\mathcal{A} = (a_{ijk}) \in \mathbb{R}_+^{[3,3]}$, where $a_{122} = 1, a_{223} = 1, a_{213} = 1, a_{322} = 1$, the remaining elements are zero.*

The majorization and representation matrices of $\mathcal{A}$ are respectively

$$M(\mathcal{A}) = \begin{pmatrix} 0 & 1 & 0 \\ 0 & 0 & 0 \\ 0 & 1 & 0 \end{pmatrix}, G(\mathcal{A}) = \begin{pmatrix} 0 & 2 & 0 \\ 1 & 1 & 2 \\ 0 & 2 & 0 \end{pmatrix}.$$

Clearly, by Definition 2.3, we know that $\mathcal{A}$ is not an essentially positive tensor or a weakly positive tensor. The linear convergence result of the NQZ algorithm cannot be derived from the literature [12,14]. Additionally, the tensor $\mathcal{A}$ is neither a generalized weakly positive tensor nor an indirectly positive tensor, which means it does not meet the conditions for linear convergence outlined in [15,16]. But by $\max\limits_{j \in \{1,3\}} r_{j2}^{\min}(\mathbb{G}(\mathcal{A})) = 1$, we know that $r_0 = 1$, and that there is $e_{22} \in \mathbb{G}(\mathcal{A})$. Thus, the NQZ algorithm is linearly convergent by Theorem 3.2.

**Example 3.2.** *Let $\mathcal{A} = (a_{i_1 i_2 \cdots i_m}) \in \mathbb{R}_+^{[m,n]}$, $m \geq 3$, where $a_{i \cdots i i+1} = (2i+1)/n, i = 1, 2, \cdots, n-1, a_{n \cdots n1} = 1$, for a certain $j_0 \in \langle n \rangle, a_{i \cdots i j_0} = (i + j_0)/n, i = 2, 3, \cdots, n$, the remaining elements are zero.*

If we take $j_0 = 1$, then the optimization matrix $M(\mathcal{A}) = 0$, and the representation matrix has the following form

$$G(\mathcal{A}) = \begin{pmatrix} * & * & 0 & \cdots & 0 & 0 \\ * & * & * & \cdots & 0 & 0 \\ \vdots & \vdots & \ddots & & \ddots & \vdots \\ * & 0 & 0 & \cdots & * & * \\ * & 0 & 0 & & 0 & * \end{pmatrix}_{n \times n},$$

where * represents non-zero elements. Clearly, by Definition 2.3, we know that $\mathcal{A}$ is not an essentially positive tensor or a weakly positive tensor. The linear convergence result of the NQZ algorithm cannot be derived from the literature [12,14]. But by $\max\limits_{i \in \langle n \rangle \backslash \{1\}} r_{i1}^{\min}(\mathbb{G}(\mathcal{A})) = 1$, we know that $r_0 = 1$, and that there is $e_{11} \in \mathbb{G}(\mathcal{A})$. Thus, the NQZ algorithm is linearly convergent by Theorem 3.2.

In Example 3.2, with $m = 3, n = 5, 10, 15$, and $j_0 = 1$, the NQZ algorithm is applied to compute the H-spectral radius in Examples 3.1 and 3.2, and its linear convergence is shown in Fig 1.

## 4 Conclusion

In this paper, we explore the linear convergence of the NQZ algorithm for calculating the $H$-spectral radius of a nonnegative tensor. By utilizing the directed graph of a tensor, we demonstrate that the NQZ algorithm exhibits $R$-linear convergence for a specific class of weakly irreducible nonnegative tensors (Theorem 3.1). We establish an upper bound for the root convergence factor $R$ and provide a general condition for the linear convergence of the NQZ algorithm (Theorem 3.2).

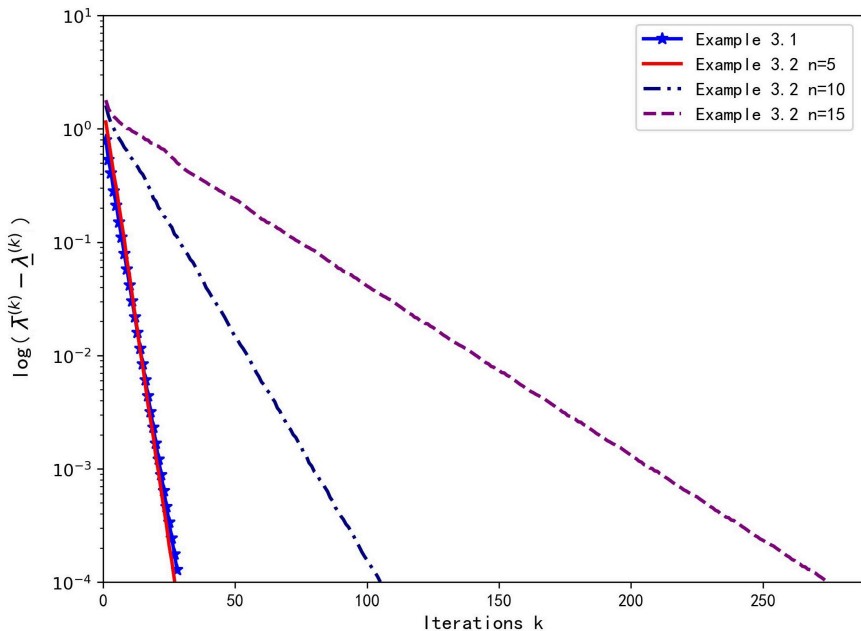

**Fig 1**. **Linear Convergence of the NQZ Algorithm for Examples 3.1 and 3.2.**

## Acknowledgments

The authors are very grateful to the reviewers for their valuable comments that improved the manuscript.

## Author contributions

**Conceptualization:** Hongbin Lv.

**Methodology:** Hongbin Lv.

**Writing – original draft:** Hongbin Lv.

**Writing – review & editing:** Hongbin Lv, Meixiang Chen.

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
