## [Decision Letter · Decision Letter 0]

23 Jul 2025

PONE-D-25-25102Linear Convergence of the NQZ Algorithm for Finding the H-spectral Radius of Nonnegative TensorsPLOS ONE

Dear Dr. Lv,

Thank you for submitting your manuscript to PLOS ONE. After careful consideration, we feel that it has merit but does not fully meet PLOS ONE’s publication criteria as it currently stands. Therefore, we invite you to submit a revised version of the manuscript that addresses the points raised during the review process.

We look forward to receiving your revised manuscript.

Kind regards,

Wen Li

Academic Editor

PLOS ONE

Journal Requirements:

This research was funded  by the National Natural Science Foundation of China No. 61772292,  the National Natural Science Foundation of Fujian Province No.2023J01997 and the Putian City Science and Technology Project No. 2022SZ3001ptxy05.

6. We note that your Data Availability Statement is currently as follows: All relevant data are within the manuscript and its Supporting Information files.

The values behind the means, standard deviations and other measures reported;

The values used to build graphs;

The points extracted from images for analysis.

7. Thank you for stating the following in the Acknowledgments Section of your manuscript: This research was funded by the National Natural Science Foundation of China No. 61772292, the National Natural Science Foundation of Fujian Province No.2023J01997 and the Putian City Science and Technology Project No. 2022SZ3001ptxy05.

This research was funded by the National Natural Science Foundation of China No. 61772292, the National Natural Science Foundation of Fujian Province No.2023J01997 and the Putian City Science and Technology Project No. 2022SZ3001ptxy05.

Additional Editor Comments:

This paper investigates the convergence properties of the NQZ algorithm for computing the H-spectral radius of nonnegative tensors. While the theoretical analysis is technically sound, the work would benefit from demonstrating practical applications or numerical experiments to illustrate the effectiveness of the proposed method. This manuscript is more suitable for researchers in linear and multilinear algebra. Therefore, I suggest the author submit it to some theoretical journals such as Linear Algebra and Its Applications or Linear & Multilinear Algebra.

Reviewers' comments:

Reviewer's Responses to Questions

**Comments to the Author**

1. Is the manuscript technically sound, and do the data support the conclusions?

Reviewer #1: Partly

Reviewer #2: Yes

2. Has the statistical analysis been performed appropriately and rigorously? 

Reviewer #1: Yes

Reviewer #2: Yes

3. Have the authors made all data underlying the findings in their manuscript fully available?

Reviewer #1: Yes

Reviewer #2: Yes

4. Is the manuscript presented in an intelligible fashion and written in standard English?

Reviewer #1: Yes

Reviewer #2: Yes

5. Review Comments to the Author

Reviewer #1: This study establishes the R-linear convergence of the NQZ algorithm for certain weakly

irreducible nonnegative tensors using directed graphs. However, the organization and structure of the manuscript, as well as several other aspects, require attentions, as detailed in the attachment.

Reviewer #2: In this paper, the authors establish the R-linear convergence of the NQZ algorithm

by using the structural properties of directed graphs of tensors. By utilizing the

directed graph of a tensor, the authors demonstrate that the NQZ algorithm exhibits Rlinear convergence for a specific class of weakly irreducible nonnegative tensors. Moreover, they establish an upper bound for the root convergence factor R and provide a general condition for the linear convergence of the NQZ algorithm. Overall, this paper is interesting.

The main results of this paper is Theorems 3.1 and 3.3. My main question is the condition used in the results. For instance, in Theorem 3.1, it is required that $e_{ii}\in G(A)$ for all $i$. Is it practical for use? Please intepret it, at least intuitively. The second question is, how to compute or estimate $r_0$ in Thm 3.1? Third, are the bounds given in this paper sharp? Maybe an eample for this is interesting.

6. PLOS authors have the option to publish the peer review history of their article (what does this mean?). If published, this will include your full peer review and any attached files.

Reviewer #1: No

Reviewer #2: No

---

## [Author Response · Author response to Decision Letter 1]

17 Sep 2025

Dear editor,

Thank you very much for your email and the constructive feedback on our manuscript "Linear Convergence of the NQZ Algorithm for Finding the H-spectral Radius of Nonnegative Tensors". We sincerely appreciate the time and effort the academic editor and reviewers have devoted to evaluating our work.

We fully understand the points raised and will carefully address each of them. We now submite the revised manuscript , it include:

A detailed rebuttal letter responding to each comment from the academic editor and reviewers, uploaded as "Response to Reviewers".

A marked-up manuscript with track changes to highlight modifications.

A clean, unmarked version of the revised manuscript, labeled "origin draft".

If there are any other question, please contact me.

Thank you again for the opportunity to revise and improve our manuscript. We look forward to your further feedback.

Best regards,

Meixiang Chen

---

## [Decision Letter · Decision Letter 1]

26 Oct 2025

PONE-D-25-25102R1

Linear Convergence of the NQZ Algorithm for Finding the H-spectral Radius of Nonnegative Tensors

PLOS ONE

Dear Dr. Chen,

Thank you for submitting your manuscript to PLOS ONE. After careful consideration, we feel that it has merit but does not fully meet PLOS ONE’s publication criteria as it currently stands. Therefore, we invite you to submit a revised version of the manuscript that addresses the points raised during the review process.

We look forward to receiving your revised manuscript.

Kind regards,

Wen Li

Academic Editor

PLOS ONE

Journal Requirements:

Additional Editor Comments:

Based on the reviewers' comments, I suggest that the author make minor revisions to the article.

Reviewers' comments:

Reviewer's Responses to Questions

**Comments to the Author**

1. If the authors have adequately addressed your comments raised in a previous round of review and you feel that this manuscript is now acceptable for publication, you may indicate that here to bypass the “Comments to the Author” section, enter your conflict of interest statement in the “Confidential to Editor” section, and submit your "Accept" recommendation.

Reviewer #1: (No Response)

Reviewer #2: All comments have been addressed

2. Is the manuscript technically sound, and do the data support the conclusions?

Reviewer #1: Yes

Reviewer #2: Yes

3. Has the statistical analysis been performed appropriately and rigorously? 

Reviewer #1: (No Response)

Reviewer #2: Yes

4. Have the authors made all data underlying the findings in their manuscript fully available?

Reviewer #1: Yes

Reviewer #2: Yes

5. Is the manuscript presented in an intelligible fashion and written in standard English?

Reviewer #1: Yes

Reviewer #2: Yes

6. Review Comments to the Author

Reviewer #1: This study establishes the R-linear convergence of the NQZ algorithm for certain weakly irreducible nonnegative tensors using directed graphs. The authors have addressed some concerns from earlier feedback, but some issues still require attention.

See the attachment for the general comments.

Reviewer #2: The authors considered my questions. I would like to recommend publishing this paper in this current form.

7. PLOS authors have the option to publish the peer review history of their article (what does this mean?). If published, this will include your full peer review and any attached files.

Reviewer #1: No

Reviewer #2: No

---

## [Author Response · Author response to Decision Letter 2]

18 Nov 2025

Dear Reviewer:

Thank you for your letter and comments concerning our manuscript entitled “Linear Convergence of the NQZ Algorithm for Finding the $H$-spectral Radius of Nonnegative Tensors” (PONE-D-25-25102). Those comments are all valuable and very helpful for revising and improving our paper.

After checking, Lemma 3.1 is the Theorem 2.4 in Reference [3], and it is confirmed that $\bar{\lambda }\left( k \right)$is indeed monotonically decreasing, which has been modified in the paper. Secondly, regarding the linear convergence of the NQZ algorithm in Lemmas 3.1 and 3.2, we plotted Figure 1 for verification. In addition, we added a paragraph explaining Figure 1 at the end of Section 3.

Please review the revised version of the paper for details.

Thank you.

---

## [Editor Report · Decision Letter 2]

25 Nov 2025

Linear Convergence of the NQZ Algorithm for Finding the H-spectral Radius of Nonnegative Tensors

PONE-D-25-25102R2

Dear Dr. Chen,

We’re pleased to inform you that your manuscript has been judged scientifically suitable for publication and will be formally accepted for publication once it meets all outstanding technical requirements.

Kind regards,

Wen Li

Academic Editor

PLOS ONE

---

## [Editor Report · Acceptance letter]

PONE-D-25-25102R2

PLOS One

Dear Dr. Chen,

I'm pleased to inform you that your manuscript has been deemed suitable for publication in PLOS One. Congratulations! Your manuscript is now being handed over to our production team.

Kind regards,

on behalf of

Dr. Wen Li

Academic Editor

PLOS One